miR-9-5p promotes myogenic differentiation via the Dlx3/Myf5 axis

Dong Liying 1 2 3 4
Wang Meng 1
Gao Xiaolei 4
Zheng Xuan 4
Zhang Yixin 4
Sun Liangjie 4
Zhao Na nazhao@hsdm.harvard.edu 5 6
Ding Chong idtbbsb@163.com 1
Ma Zeyun 7
Wang Yixiang 1 4
1 Central Laboratory, Peking University School and Hospital of Stomatology , Beijing , China
2 National Engineering Laboratory for Digital and Material Technology of Stomatology, Peking University School and Hospital of Stomatology , Beijing , China
3 Beijing Key Laboratory of Digital Stomatology, Peking University School and Hospital of Stomatology , Beijing , China
4 Department of Oral and Maxillofacial Surgery, Peking University School and Hospital of Stomatology , Beijing , China
5 Department of Restorative Dentistry and Biomaterials Sciences, Harvard School of Dental Medicine , Boston , Massachusetts , USA
6 Shanghai Stomatological Hospital, Fudan University , Shanghai , China
7 Department of VIP Service, Peking University School and Hospital of Stomatology , Beijing , China
Vassetzky Yegor
Electronic publication date: 2022 May 3
Publication date: 2022
Volume: 10
Electronic Location ID: e13360
Received 2021 Nov 29; Accepted 2022 Apr 8
Copyright: ©2022 Dong et al.
Copyright year: 2022
Copyright holder: Dong et al.
License: This is an open access article distributed under the terms of the Creative Commons Attribution License, which permits unrestricted use, distribution, reproduction and adaptation in any medium and for any purpose provided that it is properly attributed. For attribution, the original author(s), title, publication source (PeerJ) and either DOI or URL of the article must be cited.
License URL: https://creativecommons.org/licenses/by/4.0/

Keywords: MiR-9-5p, Dlx3, Myf5, Myogenic differentiation

Funding: Research Grants from Natural Science Foundation of Beijing Municipality 7182181 National Nature Science Foundation of China 81772873 81977920 81900983 Shanghai Science and Technology Young Talents Sailing Program 19YF1442500 The study was supported by Research Grants from Natural Science Foundation of Beijing Municipality (grant No. 7182181) and National Nature Science Foundation of China (grant Nos. 81772873, 81977920, 81900983) and Shanghai Science and Technology Young Talents Sailing Program (19YF1442500). The funders had no role in study design, data collection and analysis, decision to publish, or preparation of the manuscript.

==============================
MicroRNAs play an important role in myogenic differentiation, they bind to target genes and regulate muscle formation. We previously found that miR-9-5p, which is related to bone formation, was increased over time during the process of myogenic differentiation. However, the mechanism by which miR-9-5p regulates myogenic differentiation remains largely unknown. In the present study, we first examined myotube formation and miR-9-5p, myogenesis-related genes including Dlx3, Myod1, Mef2c, Desmin, MyoG and Myf5 expression under myogenic induction. Then, we detected the expression of myogenic transcription factors after overexpression or knockdown of miR-9-5p or Dlx3 in the mouse premyoblast cell line C2C12 by qPCR, western blot and myotube formation under myogenic induction. A luciferase assay was performed to confirm the regulatory relationships between not only miR-9-5p and Dlx3 but also Dlx3 and its downstream gene, Myf5, which is an essential transcription factor of myogenic differentiation. The results showed that miR-9-5p promoted myogenic differentiation by increasing myogenic transcription factor expression and promoting myotube formation, but Dlx3 exerted the opposite effect. Moreover, the luciferase assay showed that miR-9-5p bound to the 3’UTR of Dlx3 and downregulated Dlx3 expression. Dlx3 in turn suppressed Myf5 expression by binding to the Myf5 promoter, ultimately inhibiting the process of myogenic differentiation. In conclusion, the miR-9-5p/Dlx3/Myf5 axis is a novel pathway for the regulation of myogenic differentiation, and can be a potential target to treat the diseases related to muscle dysfunction.

Introduction

MicroRNAs (miRNAs) are small noncoding RNAs that inhibit translation and destabilize mRNAs (Bartel, 2009). They are involved in various biological processes and diseases as posttranscriptional gene regulators (Bushati & Cohen, 2007). Previous studies have indicated that miRNAs can regulate skeletal muscle development (Li et al., 2018; Zhang et al., 2018), cardiac regeneration (Eulalio et al., 2012), aging and functions (Boon et al., 2013). According to the function of miRNAs during myoblast differentiation, scientists named several miRNAs myomiRs (myo+miRs) to emphasize their close connection with muscle formation (Walden et al., 2009).

MyomiRs are a class of microRNAs that are considered muscle-specific or muscle-enriched and promote cell myoblast differentiation (Li et al., 2008; Siracusa, Koulmann & Banzet, 2018). In addition, other miRNAs, such as miR9-5p, could influence myogenic differentiation but those effects have not yet been explored. miR-9-5p has been shown to inhibit osteoblast proliferation and differentiation (Sun, Leung & Lu, 2016), as well as to regulate osteosarcomatous genesis (Yao, Ni & Liu, 2020); that is, miR-9-5p could influence the development of muscle and bone. Furthermore, miR-9-5p has also been found to be connected with cell adhesion (Sun, Leung & Lu, 2016). All these studies suggest that miR-9-5p could be related to myogenic differentiation. However, the exact function and underlying mechanism of how miR-9-5p regulates myoblast differentiation needs to be further studied.

Dlx3 (distal-less homeobox 3) is a homeobox gene of the distal-less family, which consists of DLX1-7 (Selski et al., 1993). During the development of vertebrate animals, DLx3 plays a crucial role by regulating the epidermis (Palazzo et al., 2017), bone (Isaac et al., 2014), tooth (Zheng et al., 2020) , and placenta development (Lichtner et al., 2013). Dlx3 has been verified to be closely related to osteogenic differentiation that controls bone mineral homeostasis (Ghoul-Mazgar et al., 2005), and is regulated by miRNAs, such as miR-124 (Zhao et al., 2016) and miR-675 (Zhao et al., 2017). It is reported that Dlx3 has the ability to alter the balance between myogenic differentiation and osteogenic differentiation (Choi et al., 2008). Furthermore, DLx3 has been linked to ectopic calcification by suppressing Runx2 expression by binding to its promoters and then regulating osteogenic differentiation and myogenic differentiation in the opposite direction (Hassan et al., 2006). Based on the above studies, we hypothesized that Dlx3 could regulate myoblast differentiation.

Myogenic factor 5 (Myf5) belongs to the myogenic regulatory factor family. The other transcription factors are Myod, Myogenin and Mrf4 (Zammit, 2017). Myf5 can influence the skeletal muscle lineage and regulate myogenic differentiation during development.

In our study, we demonstrated that miR-9-5p downregulated DLx3 and that DLx3 inhibited myogenic differentiation by directly binding to the promoter of Myf5, an important factor during myogenic differentiation. Therefore, the miR-9-5p/Dlx3/Myf5 axis could be a novel target for controlling the process of myogenic differentiation and treating myogenic diseases.

Materials and Methods

Cell culture and myogenic differentiation

The primary mouse myoblasts were isolated from limbs of 2 day old mice as described previously (Lee et al., 2015). The protocol was approved by The Institutional Animal Care and Use Committee at Peking University Health Center (LA2019-319). The mouse myoblast-derived C2C12 cell line was a gift from the National Collection of Authenticated Cell Cultures (China). C2C12 cells were cultured with Dulbecco’s Modified Eagle’s Medium (DMEM, Gibco, Carlsbad, CA, USA) supplemented with 10% fetal bovine serum (FBS, Gibco, USA), 100 U/mL penicillin and 100 µg/mL streptomycin (hereafter named the growth medium). For myogenic differentiation, the growth medium was changed to myogenic medium containing 2% horse serum instead of 10% FBS, and the other media were called the growth medium. Briefly, C2C12 cells were incubated in the growth medium described above until confluence reached 90%, followed by further incubation in myogenic medium and cultured at 37 °C in a 5% CO2 incubator. qPCR, western blotting and myotube formation assays were used to verify the level of myoblast differentiation.

Transient transfection

We seeded C2C12 cells at 1 ×105 cells per well into 6-well plates with 2 mL growth medium. Then, we transfected empty vector and OE-DLx3 plasmids into cells using Lipo8000 reagent (Beyotime, Shanghai, China) according to the manufacturer’s instructions. Transient transfections of miR-9-5p mimics or their hairpin inhibitors (RiboBio, Guangzhou, China) were performed using Lipo8000 reagent following the manufacturer’s instructions. After incubation overnight, we changed the fresh myogenic medium. At 48 h after incubation in myogenic medium, we harvested cells to detect mRNA and protein expression by qPCR and western blot.

Reverse transcription and quantitative polymerase chain reaction (qPCR)

We used TRIzol® reagent (Invitrogen Life Technologies, Grand Island, NY, USA) to isolate total RNA. RNA (2.5 µg) was reverse-transcribed into cDNA using the Superscript first-strand synthesis system (Promega, Madison, WI, USA), and 1 µg RNA was reverse-transcribed using the miDETECT A Track™ miRNA qPCR kit (RiboBio, Guangzhou, China) following the manufacturers’ instructions. qPCR was carried out as previously described (Han et al., 2019). U6 and RPS18 served as miRNA and mRNA endogenous controls, respectively. The data were analyzed by the 2−ΔΔCt method. Primers were designed as shown in Table 1.

Western blot analysis

We used RIPA buffer (Beyotime, Beijing, China) with protease inhibitors (Roche Diagnostics) to harvest protein. 30 µg protein from each sample was separated by SDS-PAGE gel (Biotides, Beijing, China), transferred to a polyvinylidene difluoride membrane, and blocked for 1 h in 5% nonfat milk at room temperature. After incubation with antibodies against DLX3 (1:1000 dilution, Abcam, Cambridge, UK), Myf5 (1:1000 dilution, Abclonal, Wuhan, China), Myod1 (1:1000 dilution, Abclonal, Wuhan, China) and RPS18 (1:1000 dilution, Abclonal, Wuhan, China) overnight at 4 °C, the membranes were incubated with HRP-conjugated secondary antibodies (1:10000, Huaxingbio, Beijing, China) for 1 h and incubated with BeyoECL Plus (Beyotime, Shanghai, China). The western blot results were quantified by ImageJ software.

Table 1 The sequences of qPCR primer used in this study.

Genes	Forward primer	Reverse primer	
Dlx3	CACTGACCTGGGCTATTACAGC	GAGATTGAACTGGTGGTGGTAG	
Myf5	AAGGCTCCTGTATCCCCTCAC	TGACCTTCTTCAGGCGTCTAC	
Myod1	CCACTCCGGGACATAGACTTG	AAAAGCGCAGGTCTGGTGAG	
MyoG	GAGACATCCCCCTATTTCTACCA	GCTCAGTCCGCTCATAGCC	
Rps18	AGTTCCAGCACATTTTGCGAG	TCATCCTCCGTGAGTTCTCCA	

Immunofluorescence staining

Cells were fixed with 4% paraformaldehyde (KeyGen Biotech, Nanjing, China) for 15 min at room temperature after incubation in myogenic medium 6 days after infection. Then, the cells were permeabilized with 0.1% Triton X-100 (Beyotime, Shanghai, China) for 20 min and blocked with 5% goat serum (Zsbio, Beijing, China) for 1 h. After incubation with the primary antibody against Desmin (1:200 dilution, Abclonal, Wuhan, China) overnight at 4 °C, the cells were washed with PBS three times and incubated with the secondary antibody (1:400 dilution, Alexa Fluor 594 anti-mouse IgG, Zsbio, Beijing, China) for 1 h at 37 °C. Finally, the cells stained with fluorescent mounting medium with DAPI (Zsbio, Beijing, China), and fluorescence micrographs were taken by fluorescence microscopy.

Plasmid construction and luciferase reporter assays

The fragment containing the miR-9-5p target site in the 3′UTR of Dlx3 mRNA was subcloned into pGL3B downstream of the luciferase gene to generate pGL3B-Dlx3-wt. Meanwhile, a mutant construct of the Dlx3 3′UTR (pGL3B-Dlx3-mut) was generated through the replacement of miR-9-5p binding site “ACCAAAGA” by “TGGTTTCT”.

Luciferase assays were performed according to the manual of the Dual Luciferase Reporter Gene Assay Kit (YEASEN, Shanghai, China). We cotransfected pGL3B empty vector, pGL3B-Dlx3-wt or pGL3B-Dlx3-mut and 100 nM miR-9-5p mimics or mimic control oligonucleotides (NC) (RiBoBio, Guangzhou, China) with the pTK-RL plasmid carrying the Renilla luciferase expression cassette into 293T cells, respectively, using Lipo8000 reagent (Beyotime, Shanghai, China). At 24 h posttransfection, the supernatant of the lysed cells was harvested to perform a luciferase assay according to the manufacturer’s manual.

Rescue assay

siRNA sets specific to mouse Myf5 and Dlx3 were synthesized by RiboBio Co., Ltd. (Guangzhou, China), respectively. C2C12 cells were cotransfected with siMyf5 or control (siNC) and siDlx3 and then incubated for 24 h. Cells were then harvested and analyzed using qPCR and western blot assays.

Statistical analysis

All data are presented as the means ± standard deviation (SD) and were analyzed with GraphPad Prism v6.0 software. For two group comparisons, an unpaired t test was used. For multiple group comparisons, one-way ANOVA analysis was used, followed by Tukey’s post-hoc test. Differences were considered statistically significant when P < 0.05. All experiments were performed at least three times.

Results

Dlx3 is downregulated, whereas miR-9-5p is upregulated in the process of myogenic differentiation

To identify the expression profile of Dlx3 and miR-9-5p during myogenic differentiation, primary mouse myoblasts were incubated in myogenic medium for 0, 2, 4 and 6 days, and then expression levels were detected by qPCR. Clearly, Myod1, Myf5, Mef2c, Desmin and MyoG were increased in a time-dependent manner. Compared with those in the initial state, the expression level of Dlx3 was decreased on Days 2, 4 and 6, while miR-9-5p was increased at the initial state and decreased at Day 6, but still higher than Day 0 (Fig. 1A). Then C2C12 cells were used to do the next experiments. C2C12 cells were incubated in myogenic medium for 0, 2, 4 and 6 days, and then expression levels were detected by qPCR and western blot analyses. Under a microscope, we found that the number of myotubes was increased during the progression of myogenic differentiation in C2C12 cells (Fig. 1B). After incubation in differentiation medium for 6 days, the myotubes could self-contract (Video S1). During the process of myogenic differentiation, compared with those in the initial state, the expression level of Dlx3 was decreased on Days 2, 4 and 6 (Fig. 1C), while miR-9-5p was increased on Days 2 and then decreased on Days 4 and 6, but still higher than Day 0 (Fig. 1C). In addition, Myod1, Myf5, Mef2c, Desmin and MyoG were clearly increased in a time-dependent manner (Fig. 1D), which was consistent with the western blot results (Fig. 1E). These data indicated that increased miR-9-5p expression and decreased Dlx3 expression participate in myogenic differentiation.

Figure 1 Myogenic induction downregulates Dlx3 and upregulates miR-9-5p expression.

(A) The expression of myogenic differentiation related genes as well as Dlx3 and miR-9-5p were detected by qPCR after primary mouse myoblast cells were induced in myogenic medium at the indicated days. The internal controls were Rps18 for mRNA and U6 for miR-9-5p, respectively. (B) Myotubes formation was observed under microscope, Bar = 200 µm for 4× images, and Bar = 40 µm for 20× images. (C–E) The expression of miR-9-5p, Dlx3 and myogenic differentiation related genes mRNAs were confirmed by qPCR (C & D) and western blot (E) in C2C12 cells, respectively. The numbers above each band of western blot were quantified by ImageJ software and represent the relative expression level of each protein in different groups. RPS18 protein served as the loading control. (∗)P < 0.05, (∗∗)P < 0.01, (∗∗ ∗)P < 0.001, (∗∗ ∗ ∗)P < 0.0001.

miR-9-5p promotes myotube formation via downregulation of Dlx3 expression

To explore the underlying mechanism by which miR-9-5p regulates myogenic differentiation, miR-9-5p mimics and miR-9-5p inhibitor were transfected into C2C12 cells. As shown in Fig. 2A, miR-9-5p was increased in the miR-9-5p mimics group, while the decreased expression was examined in the miR-9-5p inhibitor group. The expression of Dlx3 was decreased and Myod1 and Myf5 were increased in the miR-9-5p mimics group. Conversely, in the miR-9-5p inhibitor group, Dlx3 was increased and Myod1 and Myf5 were decreased (Fig. 2A). The results were further confirmed by western blot (Fig. 2B). Meanwhile, we evaluated myotube formation. The myotube numbers were significantly increased in the miR-9-5p mimics group, and clearly decreased in the miR-9-5p inhibitor group, which suggested that miR-9-5p enhanced myotube formation (Fig. 2C). According to the results of the immunofluorescence staining of Desmin, miR-9-5p mimics promoted myogenic differentiation, and miR-9-5p inhibitor suppressed myogenic differentiation (Fig. 2D).

Figure 2 MiR-9-5p negatively regulates Dlx3 expression via directly targeting Dlx3 3′UTR.

(A & B) C2C12 cells were transiently transfected with mimics (50 nM) or inhibitor miR-9-5p (100 nM) and their negative control, respectively, then incubated in myogenic medium for 2 days and examined by qPCR (A), and western blot analysis (B). (C) After 6 days, the numbers of myotubes per field were counted from four randomly chosen fields under microscope. Bar = 245 µm. (D) Immunofluorescence staining of Desmin in C2C12 cells after overexpression or knockdown of miR-9-5p. Bar = 200 µm. (E) The luciferase report plasmids were constructed as (E top panel) in pGL3B vector. Luciferase assay verifies that DLX3 is the target of miR-9-5p (E bottom panel). (∗)P < 0.05, (∗∗)P < 0.01, (∗∗ ∗)P < 0.001.

Next, we focused on the miR-9-5p negative regulatory effect on Dlx3. We investigated whether Dlx3 was the target gene of miR-9-5p and whether miR-9-5p could downregulate Dlx3 by binding to its 3′UTR. First, we used miRBD (http://mirdb.org/) and TargetScan (http://www.targetscan.org/) to predict supposed miRNA binding sites. Both databases predicted a binding site for miR-9-5p in the Dlx3 3′-UTR. To further validate the prediction, we constructed plasmids that contained sequences of wild-type (WT) or mutant (MUT) Dlx3 3′-UTR and performed a luciferase reporter assay. The results showed that mutant Dlx3 could rescue the reduction in WT-Dlx3 binding with miR-9-5p mimics, which confirmed that miR-9-5p could bind to the 3′UTR of Dlx3, and inhibit its transcriptional expression (Fig. 2E). The results indicated that miR-9-5p promoted myotube formation by targeting the Dlx3 3′UTR and then negatively regulated Dlx3 expression.

Overexpresson of Dlx3 inhibited myogenic differentiation, whereas knockdown of Dlx3 had the opposite effect

To further explore the role of Dlx3 in myogenic differentiation, C2C12 cells were transiently transfected with Dlx3 overexpression (OE-Dlx3) and Dlx3 siRNA (siDlx3) plasmids. As shown in Figs. 3A and 3B, the qPCR and western blot results showed that Dlx3 inhibited myogenic differentiation at both the mRNA and protein levels. Overexpression of Dlx3 decreased the expression levels of myogenic marker genes, such as Myod1 and Myf5, compared with those in cells transfected with empty vector (EV). Furthermore, Dlx3 overexpression significantly decreased the numbers of total myotubes (Fig. 3C).

Figure 3 Dlx3 inhibits myogenic differentiation.

(A–C) C2C12 cells were transiently transfected with Dlx3 overexpression plasmid (OE-Dlx3) and empty vector (EV), respectively, then incubated in myogenic medium for 2 days and Dlx3, Myod1 and Myf5 expression was examined by real-time PCR (A), and western blot (B). 6 days after myogenic induction, the numbers of myotubes per field were counted from four randomly chosen fields under a microscope (C). Bar = 100 µm. (D-F) C2C12 cells were transiently transfected with siDlx3 and siRNA negative control (siNC), respectively, then followed the above assays using qPCR (D), western blot (E) and myotube formation (F). (G) Immunofluorescence staining of desmin in C2C12 cells transfected OE-Dlx3, siDlx3 and their controls. Bar = 100 µm. (∗)P < 0.05, (∗∗)P < 0.01, (∗∗ ∗)P < 0.001.

Then, we induced siDlx3 in C2C12 cells to knockdown Dlx3. The qPCR and western blot results demonstrated that both the mRNA and protein levels of Dlx3 were significantly decreased, accompanied by the upregulated expression of Myod1 and Myf5 (Figs. 3D and 3E). The numbers of total myotube numbers were increased accordingly (Fig. 3F). The results of the immunofluorescence staining of desmin indicated that Dlx3 inhibited myogenic differentiation since there were fewer myotubes in the OE-Dlx3 group than in the EV group, and more in the siDlx3 group than in the NC group (Fig. 3G). These findings confirmed that Dlx3 inhibited the process of myogenic differentiation.

Dlx3 binds to the promoter of Myf5 and negatively regulates Myf5

Next, to explore the underlying mechanism by which Dlx3 inhibits myogenic differentiation, we aimed to confirm whether Dlx3 regulated the downstream gene as a transcription factor by targeting the promoter of the downstream genes. According to the results above, we noticed that Myf5 was significantly changed when Dlx3 was overexpressed or knocked down. We performed bioinformatics analysis of Dlx3 and the promoter of Myf5 by using JASPAR (http://jaspar.genereg.net/), and the results showed that Dlx3 could bind to the promoter of Myf5 (Fig. 4A). Then, we used luciferase reporter assays and designed plasmids that contained blank, wild-type or mutant Myf5 promoter sequences. OE-Dlx3 was cotransfected with luciferase reporter plasmid and pTK-RL plasmid. A luciferase assay demonstrated that Dlx3 downregulated Myf5 by binding to the promoter of Myf5 (Fig. 4B). Therefore, Dlx3 directly downregulated Myf5 to inhibit myogenic differentiation.

Figure 4 Dlx3 downregulates Myf5 by binding to its promoter and verified by luciferase assay.

(A) Schematic diagram showing the constructed luciferase report plasmids in pGL3B vector. (B) Luciferase assay confirmed that Dlx3 regulates the expression of Myf5. (C & D) C2C12 cells were co-transfected with siDlx3, and either siMyf5 or siRNA negative control, as well as pTK-RL plasmid with Renilla luciferase expression cassette. Myogenic differentiation-related genes expression was detected by qPCR and western blot at mRNA (C) and protein (D) levels. (E) The numbers of myotubes per field were counted from four randomly chosen fields under a microscope. Bar = 200 µm. (F) Fluorescence micrographs to detect Desmin in C2C12 cells were performed. Bar = 100 µm. (∗)P < 0.05, (∗∗)P < 0.01, (∗∗ ∗)P < 0.001, (∗∗ ∗ ∗)P < 0.0001.

Furthermore, we performed a rescue assay to confirm the relationship between Dlx3 and Myf5. We cotransfected the siDLX3 and siMyf5 and examined the expression of Myod1 and Myf5 at both the mRNA and protein levels by qPCR (Fig. 4C) and western blot (Fig. 4D). The results showed that the expression levels of Myf5 and Myod1 were increased when Dlx3 was knocked down and then decreased when both Dlx3 and Myf5 were knocked down. The changes in total myotube numbers (Fig. 4E) and the results of the immunofluorescence staining of Desmin also supported the results above (Fig. 4F). These results showed that siMyf5 could rescue the increase in myoblast differentiation caused by siDlx3, which supported the conclusion that miR-9-5p upregulated myogenic differentiation via the Dlx3/Myf5 axis (Fig. 5).

Figure 5 The mechanism of miR-9-5p regulating myogenic differentiation through DLX3/Myf5 axis.

Discussion

Muscles are involved in the most vital movements of humans, such as respiration (Bye, Farkas & Roussos, 1983), heartbeat (Sonnenblick & Skelton, 1971) and exercise (Zheng et al., 2021). Dysfunctions of muscle may harm physiological activities, and even be life-threatening. Research on muscle-related diseases and drug discovery and development are of great significance.

With the discovery of noncoding RNA, an increasing number of miRNAs have been found to regulate vital activities, cause diseases and develop new treatment methods. An increasing number of miRNAs have been used as potential small molecule drugs to cure diseases, especially in alternative therapies and vaccines (Yu, Choi & Tu, 2020). miR-9-5p has been reported to participate in the regulated signaling pathway in osteosarcoma (Yao, Ni & Liu, 2020), indicating that miR-9-5p may influence osteogenic differentiation and myogenic differentiation, but the related research is still scarce. We first investigated the regulatory role of miR-9-5p in C2C12 cells during myogenic differentiation which has not been reported before. Since miR-9-5p could promote myogenic differentiation, miR-9-5p may support muscle regeneration. That is, miR-9-5p could be a potential small molecule drug to cure muscle injury related diseases. miR-9-5p may promote muscle formation to improve muscle weakness. Although the regulatory relationship between miRNAs and mRNA is not simply one-to-one, there is a time-lag between theoretical research and practical applications.

Dlx3 and other members of the distal-less family play an important role in the growth and development of vertebrate animals (Selski et al., 1993). They influence the formation of many organs and systems. Tricho-Dento-Osseous syndrome (TDO syndrome), a rare genetic disease, has been demonstrated to be caused by mutation of DLx3, with curly hair, enamel hypoplasia, dentinogenesis imperfecta, and increased density of jaw and skull bones (Lichtenstein et al., 1972). Since Dlx3 has been reported to downregulate myogenic differentiation (Choi et al., 2008) and Dlx3 is downregulated by miR-9-5p as its target gene, we attempted to verify the regulatory role of Dlx3 and miR-9-5p during myogenic differentiation. We investigated whether miR-9-5p directly suppresses Dlx3 by targeting the Dlx3 3′UTR and whether Dlx3 inhibits myogenic differentiation by directly binding to the promoter of Myf5 and downregulating Myf5 expression. Collectively, miR-9-5p promotes myogenic differentiation via the Dlx3/Myf5 axis.

In the study of bone formation, it was discovered that there might be a balance between myogenic differentiation and osteogenic differentiation and that they can transform into each other (Ying, Hussain & Zhang, 2003). Both miR-9-5p and Dlx3 can regulate osteogenic differentiation (Isaac et al., 2014; Zhao et al., 2016) and myogenic differentiation, and they may be novel research objects of ossification of muscle, which causes ankylosis after burn or trauma (Ring & Jupiter, 2004). Traumatic and joint replacements can cause heterotopic ossification, also known as intramuscular ossification. Knockout of DLx3 promoted myoblast differentiation, and DLx3 mutation increased the stemness of bone marrow mesenchymal stem cells, resulting in cumulative bone formation. With increased muscle strength, strength increases, which also promotes bone formation. These results suggest that DLx3 can regulate the amount of bone and muscle but is not a determinant of the direction of differentiation leading to osteogenic or myogenic differentiation.

Usually, we may think gene mutation is harmful to people’s health, but some may cause “advantageous phenotypes”, which may benefit health and have implications for treating diseases. According to the function of the gene, gene editing, such as mutating or knocking down the gene, could be used as a therapeutic method, but implementation is difficult because of the nondeterminacy of gene mutation and the complexity of gene regulation. There is still a long way to go to solve the problem of ethical issues and security. However, studies of gene function and gene editing therapy still provide novel ideas and the possibility to treat some diseases.

Conclusions

In conclusion, our study first demonstrated that the miR-9-5p/Dlx3/Myf5 axis is a novel pathway for the regulation of myogenic differentiation. Moreover, it is a potential target to treat the diseases related to muscle dysfunction.

Supplemental Information

Supplemental Information 1 Myotubes self-contracts after myogenic indution

Click here for additional data file.

Supplemental Information 2 Raw data

Click here for additional data file.

The authors thank all staffs in Central Laboratory, Peking University School of Stomatology for providing experimental techniques to help in this work.

Additional Information and Declarations

Competing Interests

Author Contributions

Animal Ethics

Data Availability

The authors declare there are no competing interests.

Liying Dong performed the experiments, analyzed the data, prepared figures and/or tables, authored or reviewed drafts of the paper, and approved the final draft.

Meng Wang performed the experiments, authored or reviewed drafts of the paper, and approved the final draft.

Xiaolei Gao, Xuan Zheng and Yixin Zhang performed the experiments, analyzed the data, authored or reviewed drafts of the paper, and approved the final draft.

Liangjie Sun analyzed the data, authored or reviewed drafts of the paper, and approved the final draft.

Na Zhao, Chong Ding, Zeyun Ma and Yixiang Wang conceived and designed the experiments, analyzed the data, authored or reviewed drafts of the paper, and approved the final draft.

The following information was supplied relating to ethical approvals (i.e., approving body and any reference numbers):

The protocol was approved by The Institutional Animal Care and Use Committee at Peking University Health Center (LA2019-319).

The following information was supplied regarding data availability:

The raw data are available in the Supplementary Files.

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
