# Peer review of "miR-9-5p promotes myogenic differentiation via the Dlx3/Myf5 axis"

_PeerJ, doi:10.7717/peerj.13360_

## Round 0.1 · original submission · Major Revisions

Please answer to the reviewers' criticisms and improve the statistical analysis.

·

Basic reporting

The manuscript entitled “miR-9-5p promotes myogenic differentiation via Dlx3/Myf5 axis” by Liying Dong et al. identified a novel pathway that miR-9-5p induced myogenic differentiation by targeting Dlx3 3-UTR to inhibit its binding to Myf5 promoter. The study is of clinical significance and provides potential therapeutic targets.

Experimental design

1. Authors well studied the miR-9-5p/DLX3/Myf5 axis in mouse myoblast-derived C2C12 cell line. Major experiments should be reproduced in a second cell line.
2. In Figure 2C, miR-9-5p mimics increased myotube numbers compared to NC mimics while miR-9-5p inhibitor decreased myotube numbers compared to NC inhibitor. Why were myotube numbers so different between two controls, NC mimics (about 20) and NC inhibitor (about 60) groups? Authors should present these four groups in one panel to compare myotube numbers.
3. In Figure 4C, siDLX3 alone inhibited expression of DLX3. Why did co-transfection with siMyf5 increase DLX3 level?
4. In Methods, Statistical analysis should be elaborated. It is too simply just stating “We used one-way ANOVA analysis or t-test to determine statistical significance”. For example, in Figure 1B and Figure 4B, C and E, the statistical method should be properly described. One-way ANOVA is used to test whether it is statistically different among all compared groups. Multiple comparison is followed to test which two groups are different.

Validity of the findings

In Figure 1B, miR-9-5p was increased on day 2 and then back to the initial level on days 4 and 6. What is the interpretation? The data does not support the conclusion that the increased miR-9-5p expression participates the myogenic differentiation.

Additional comments

None.

Reviewer 2 ·

Basic reporting

In this study, Dong et al proposed that the myogenic differentiation of C2C12 cells was regulated by miR-9-5p/Dlx3/Myf5 axis. They found Dlx3 was decreased, whereas miR-905p was increased, during myogenic differentiation. And miR-9-5p promoted myotube formation via decreasing Dlx3. In addition, the suppression of the myogenic differentiation in C2C12 cells were mediated by Dlx3, and that was associated with its capacity in reducing Myf5. This is a straightforward study. It is easy to follow. Following are the comments:
1. Language editing is needed. I would suggest the authors to seek help from a native English speaker.
2. The statistical method used in this study is improper. ANOVA only tells whether there are significant differences between all compared groups. Post-hoc test is needed to further clarify where the differences are. In addition, please specify which type of t-test was used in this study, paired or unpaired.
3. Immunofluorescence staining of muscle protein markers during indicated time points is recommended. For instance, the authors can use alpha-actinin or MHC as markers
4. The biggest drawback of this study is that the changes of protein levels in western blot were not as robust as which were described in the quantification plots. For example, the change of Myod1 was barely visible after miR-9-5p inhibitor treatment in Figure 2B. However, the authors indicated there was significant decrease in their quantification plot. I would strongly suggest the authors to reverify there western blot results. Almost every WB data has this issue. In addition, the Myf5 bands in Figure 2B are hazy.
5. According to the data, Myod1 was also changed after miR-9-5p mimic and inhibitor treatment, and after knockdown or overexpressing Dlx3. Why did the authors were more interested in Myf5?
6. I would suggest the authors to prove if modulating the expression of Myf5 alone is enough to delay or facilitate the myogenic differentiation. Otherwise, the proposed regulatory flux will be compromised.

Experimental design

Please see details in Basic reporting.

Validity of the findings

Please see details in Basic reporting.

·

Basic reporting

MiR-9-5p is a popular microRNA and well studied in human cancers, fibrosis development and other diseases. However, its function in myogenic differentiation is rarely explored. The manuscript entitled “miR-9-5p promotes myogenic differentiation via Dlx3/Myf5 axis” unveiled miR-9-5p prompts bone formation through suppress Dlx3 mRNA expression, while increases Myf5 which is an essential transcript factor of myogenic differentiation. All the experiments are well designed and convincing. However, several question should be illustrated and minor modified to improve the paper before publication.

1. In figure 1A, the bar value should be noted, so do figure2C and figure 3C&F. And in figure 1B&C should notify mRNA level fold change in Y axis.
2. Fig2B, why the western blot of Myf5 is so obscure, I couldn’t tell the difference. Please run this experiment again.
3. In fig2C, why NC mimics and NC inhibitors are so dramatically different? Is because the cells are at different time point or something else, please demonstrate this in the paper?
4. All the western blot results should be quantified.

Experimental design

no comment

Validity of the findings

no comment

Additional comments

no comments

---

## Round 0.2 · Minor Revisions

Please revise the manuscript according to the remarks of Reviewer 1

·

Basic reporting

Authors have successfully responded to my comments. My only concern is the lack of the second cell line to valid their key findings. This drawback will significantly decrease the impact and validity of the manuscript. It is a poor justification that "Most papers only used C2C12 in their studies".

Experimental design

None.

Validity of the findings

None.

Additional comments

None.

Reviewer 2 ·

Basic reporting

The authors have improved their manuscript according to my comments. If it is applicable, this manuscript can be considered for publication in this current version.

Experimental design

No comment

Validity of the findings

No comment

·

Basic reporting

All the questions are clearly answered and well revised. I do recommend this manuscript could be accepted.

Experimental design

no comment

Validity of the findings

no comment

---

## Round 0.3 · accepted · Accept

Congratulations! Your paper can now be published!